# Nutritional and physicochemical characteristics of purple sweet corn juice before and after boiling

Xuanjun Feng[1,2,3☉], Liteng Pan[1,2,3☉], Qingjun Wang[1,2,3], Zhengqiao Liao[1,2,3], Xianqiu Wang[1,2,3], Xuemei Zhang[1,2,3], Wei Guo[1,2,3], Erliang Hu[2,3], Jingwei Li[2,3], Jie Xu[1,2,3], Fengkai Wu[1,2,3], Yanli Lu[1,2,3]*

1 State Key Laboratory of Crop Gene Exploration and Utilization in Southwest China, Sichuan Agricultural University, Wenjiang, Ya'an, Sichuan, China, 2 Maize Research Institute, Sichuan Agricultural University, Wenjiang, Ya'an, Sichuan, China, 3 Key Laboratory of Biology and Genetic Improvement of Maize in Southwest Region, Ministry of Agriculture, Wenjiang, Chengdu, Sichuan, China

☉ These authors contributed equally to this work.
* yanli.lu82@hotmail.com

**Data Availability Statement:** All relevant data are within the paper and its Supporting Information files.

## Abstract

Sweet corn juice is becoming increasingly popular in China. In order to provide valuable health-related information to consumers, the nutritional and physicochemical characteristics of raw and boiled purple sweet corn juices were herein investigated. Sugars, antinutrients, total free phenols, anthocyanins, and antioxidant activity were analyzed by conventional chemical methods. The viscosity and stability of juices were determined by Ubbelohde viscosity meter and centrifugation, respectively. Boiling process could elevate viscosity, stability and sugar content, and reduce antinutrients, total free phenols, anthocyanins, and antioxidant activity in corn juice. In addition, short time boiling efficiently reduced the degradation of anthocyanins during subsequent refrigeration. The content of amino acids, vitamin B1/B2 and E were detected by High Performance Liquid Chromatography. Gas Chromatography Mass Spectrometry was used for the analysis of fatty acids and aroma compounds. Several aroma compounds not previously reported in corn were identified, including 1-heptanol, 2-methyl-2-butenal, (Z)-3-nonen-1-ol, 3-ethyl-2-methyl-1,3-hexadiene, and 2,4-bis (1,1-dimethylethyl)phenol. Interestingly, the boiling process had no apparent effect on the amino acids profile, but it caused a 45.8% loss of fatty acids in the juice by promoting the retention of fatty acids in the corn residue. These results provide detailed information that could be used for increasing consumers' knowledge of sweet corn juice, further development of sweet corn juice by food producers, and maize breeding programs.

## Introduction

Sweet corns differ from normal field corns in the mutation of starch synthesis genes, e.g., *Sugary* (*Su*), *Shrunken* (*Sh*), and *Brittle* (*Bt*), which result in a higher accumulation of sugar and water-soluble polysaccharides at the milk stage than in normal field corns, and provide the sweet taste and creamy texture of sweet corns [1]. Consequently, sweet corns have a slightly

**Funding:** YL received two fundings. The National Key Research and Development Program of China (2018YFD0100100) and Science and technology support program of Sichuan province, China (2016NYZ0029). The funders had no role in study design, data collection and analysis, decision to publish, or preparation of the manuscript.

**Competing interests:** The authors have declared that no competing interests exist.

different metabolic change in their carbohydrates, and there is a different nutritional profile compared with normal field corns. Generally, sweet corns have higher lipid and amino acid contents, especially the limiting amino acid lysine, than normal field corns at the milk stage [2–4].

In addition to good taste, sweet corn is rich in nutrients and bioactive compounds, including amino acids, unsaturated fats, dietary fiber, vitamins, minerals, phenols, and other phytochemicals [5]. Corn has the highest content of phenols among common grains, including rice, wheat, and oats [6]. The phenol content is directly related to total antioxidant activity, such that corn has the highest total antioxidant activity, which is beneficial for human health [6,7]. It was reported that consumption of corn and its derived products helps reduce the risk of chronic diseases like cardiovascular disease, type II diabetes, and obesity [5,8,9]. Therefore, the consumption of corn is advocated and has been increasing annually worldwide [7,10].

Sweet corns are one of the most popular vegetables in North America and China and are becoming increasingly popular in the rest of the world as well [5]. Canned-, frozen-, and fresh-sweet corn are major traditional forms of its consumption, and sweet corn ranks third among vegetables consumed in the United States [5]. Consumption of the fresh juice of sweet corn has increased rapidly in China in recent years because of its pleasant taste and rich nutritional content. Sweet corn juice can optionally be boiled before drinking. Boiling enhances the stability of corn juice by starch gelatinization, improves the fragrance via the release of aroma compounds, and can kill pathogenic bacteria to better meet food safety requirements. However, cooked foods have been widely considered to have lower nutritional value, such as lower phenols, anthocyanins, antioxidant, and vitamin contents than the corresponding fresh commodities [11–16]. Although heating is usually associated with nutrition loss in most foods, thermal processing has been reported to elevate the total antioxidant activity and total phenols contents in sweet corn, kale, white cabbage, and onion [11,14,15]. Boiling and roasting have been found to increase the total amino acid contents in four marine fishes [17], and microwaving increased the vitamin A and vitamin E contents in African catfish [18]. In general, boiling can effectively reduce anti-nutritional factors and improve nutritional quality, such as enhancing the true ileal digestibility of egg protein and improving tocopherol content [4,16,19,20]. Anti-nutritional factors, such as trypsin inhibitors, tannin, and phytic acid, have inhibitory effects on digesting enzymes and can retard mineral absorption, thus limiting the bioavailability of nutrients [16].

Along with the increasing consumption of sweet corn juice in China and popularity of anthocyanin-enriched foods, a purple sweet corn, which typically has higher anthocyanins than yellow or white sweet corns, was selected for comparing the nutritional and physicochemical characteristics of juices before and after boiling. Collectively, several major results were listed here. 1. The boiling process could elevate viscosity, stability, and sugar content, and reduce antinutrients, total free phenols, anthocyanins, and antioxidant activity in corn juice. 2. The boiling process caused a 45.8% loss of fatty acids in the juice by promoting the retention of fatty acids in the corn residue. 3. Heat blanching for a short time (boiling at 95˚C for 5 min) suppressed the degradation of phenols and anthocyanins, probably through an enzymatic pathway. 4. Several aroma compounds previously unreported in corn were identified in this purple sweet corn. The results of this study could provide more information for increasing consumers' knowledge of sweet corn juice, further development of sweet corn juice by the food producers, and maize breeding programs.

## Materials and methods

### Chemicals

Sodium chloride, methanol, ethanol, amino acid standards, acetonitrile, phenyl isothiocyanate, gallic acid, triethylamine, *n*-hexane, boron trifluoride-methanol, dichloromethane, formic

acid, diethyl ether, petroleum ether, sodium hydroxide, potassium hydroxide, hydrochloric acid, and perchloric acid were purchased from ANPEL Laboratory Technologies (Shanghai) Inc. Vitamin B1/B2 and vitamin E were purchased from Shanghai Yuanye Bio-Technology Co., Ltd. Folin-Ciocalteu reagent, 2,2-diphenyl-1-picrylhydrazyl (DPPH), mixed standards of fatty acids, cyanidin 3-*O*-glucoside, anthranone, 1,2,3-trihydroxybenzene, benzoyl-DL-arginine *p*-nitroanilide hydrochloride, dimethyl sulfoxide, sulfosalicylic acid, trypsin, and phytic acid were purchased from Sigma-Aldrich Chemical Co. (St. Louis, MO, USA). The purity of chemicals used for High-Performance Liquid Chromatography (HPLC) and Gas Chromatography Mass Spectroscopy (GCMS) complied with the requirement of relevant procedures. All other chemicals were of analytical grade.

## Sample preparation

The purple sweet corn (SICAU76, bred by Sichuan Agricultural University, China) used herein was cultured at a farm in Chongzhou in the summer of 2019, and four biological replicates were prepared. Fresh ears were harvested 23 days after pollination. Three biological replicates and six ears of each replicate were dehusked, and the kernels were removed from the cobs and mixed together. Then, 400 g of kernels (water content = 69.5%) were weighed and homogenized with 1200 mL water using a juice extractor at 10,000 rpm for 1 min. The homogenate was divided into two equal parts. One part was designated as the raw sample. The other part was sealed in a bottle and heated at 95˚C for 5 min to prepare the boiled sample. The raw and boiled homogenates were filtered through two layers of medical gauze (cotton fabrics with 1 mm aperture size). After filtration, a mean of 5.428 g (5.217, 5.465, and 5.602 g for each replicate) and 4.571 g (4.41, 4.521, and 4.782 g for each replicate) dry residues were obtained from the raw (residue ratio, 8.90%) and boiled (residue ratio, 7.49%) homogenates, respectively. Samples were stored at −80˚C and thawed at room temperature before use. The thermal processing used for the volatile components assay is different and is described in the corresponding section.

## Statistical analysis

All of the experiments were repeated more than three times and each time with three technical repetitions. Values are displayed as the Mean ± SD (standard deviation). Differences between the raw and boiled juice were determined by paired t-test using GraphPad Prism 7, and by two-way ANOVA using Excel 2010. Significance was determined at two levels: $P < 0.05$ and $P < 0.01$.

## Amino acid profile assay

The corn juice samples (3 mL) were hydrolyzed in a closed vessel using 6 mol/L HCl at 110˚C for 24 h. After hydrolysis, 1 mL of supernatant was evaporated to dryness at 85˚C in a water bath, and then 1 mL of water was added to the dried hydrolysates, and the mixture was evaporated to dryness again. HCl (0.02 mol/L, 10 mL) was mixed with the hydrolysates. For the derivative reaction, 500 μL of the hydrolysates was reconstituted with 250 μL phenyl isothiocyanate (0.1 mol/L in acetonitrile) and 250 μL triethylamine (1 mol/L in acetonitrile) for approximately 1 h. *n*-Hexane (2 mL) was added to the derivatives, followed by vibrating and allowing to stand until the layers separated. After stratification, the bottom layer was filtered through cellulose membrane (pore size = 0.45 μm) syringe filters.

Amino acid analysis was performed using an Agilent-1260 HPLC with a C18 chromatographic column (Shiseido, 4.6 mm × 250 mm × 5 μm). The process used was modified from the China national standard (GB/T 30987–2014). The injection volume was 10 μL, the

column temperature was 40˚C, and the mobile phase consisted of solvent A (97 vol% sodium acetate solution (0.1 mol/L) and 3 vol% acetonitrile, pH 6.5) and solvent B (80 vol% acetonitrile and 20 vol% water). The elution conditions were as follows: 0–14 min, solvent A was reduced from 100 to 85%; 14–29 min, solvent A was reduced from 85 to 66%; 29–30 min, solvent A was decreased to 0%; 30–37 min, solvent B was held at 100%; 37–38 min, solvent B was decreased to 0%; 38–45 min, solvent A was held at 100%. All the gradients were linear, and the flow rate was 1 mL/min.

The total protein content of the juice was estimated from the total amino acids content. The predicted protein efficiency ratio (P-PER) was calculated using the following equations developed by Alsmeyer, Cunningham, and Happich (1974), as cited by Mohapatra et al. [16].

$$PER = -0.684 + 0.456(Leu) - 0.047(Pro) \tag{1}$$

$$PER = -0.468 + 0.454(Leu) - 0.105(Tyr) \tag{2}$$

## Fatty acid profile assay

The hydrolysis was performed as follows: 3 mL of corn juice or 1 g corn residue, 2 mL 95% ethyl alcohol, 100 mg pyrogallic acid, and several zeolites were added to a flask. Then, 10 mL HCl (6 mol/L) was added and the temperature was maintained at 80˚C for 40 min. The lipid extraction was performed as follows: ethyl alcohol (95%, 10 mL) was mixed with the hydrolysate and the mixture was transferred to a separatory funnel. Then, 50 mL ether-petroleum ether (50:50 vol%) was used to rinse the flask and subsequently transferred to the separatory funnel. The mixture was vibrated for 5 min in the separatory funnel and allowed to stand for 10 min. The organic layer was collected in a new flask and the process was repeated three times. Finally, the mixture was evaporated in a water bath and dried in an oven at 100˚C for 2 h.

Esterification was performed as follows: 2 mL 2% NaOH/methyl alcohol solution (2 g NaOH dissolved in 100 mL methyl alcohol) was added to the dried extraction and kept at 85˚C for 30 min. Next, 3 mL 14% boron trifluoride (BF$_3$)/methyl alcohol solution (14 g boron trifluoride dissolved in 100 mL methyl alcohol) was added and maintained at 85˚C for 30 min. Then, 1 mL *n*-hexane was added, the mixture was vibrated for 2 min, and then allowed to stand for 1 h. A 100-μL aliquot of the supernatant liquid was collected and diluted to 1 mL using *n*-hexane. Finally, the solution was filtered through cellulose membrane (pore size = 0.45 μm) syringe filters.

The fatty acid profile was determined by GCMS using a Thermo Trace1310/ISQ instrument with a TG-5MS (30 m × 0.25 mm × 0.25 μm) chromatographic column. The applied process was modified from the China national standard (GB 5009.168–2016). The temperature program was as follows: 80˚C for 1 min and heating to 200˚C by 10˚C/min; 5˚C/min to 250˚C, 2˚C/min to 270˚C, and finally holding at 270˚C for 3 min. The instrument parameters were: inlet temperature = 290˚C; carrier gas flow rate = 1.2 mL/min; split ratio = unsplit; ion source temperature = 280˚C; transfer line temperature = 280˚C; solvent delay time = 5 min; scanned area = 30–400 amu; ion source = EI, 70 eV.

## Identification of volatile components

Raw corn juice (1 mL) was added to a headspace bottle and mixed with 3 mL of saturated NaCl. The volatile components isolated after 30 min at 37˚C and 95˚C were considered to come from the raw and boiled juices, respectively.

The volatile components were detected by GCMS using an Agilent 6890N-5975B with a solid-phase 100 μL polydimethylsiloxane microextraction fiber and an HP-5MS (30 m × 0.25

mm × 0.25 μm) chromatographic column with a carrier gas flow rate of 1 mL/min. The applied process was modified from the China national standard (GB/T 35862–2018). The temperature program was as follows: 45˚C for 4 min, heating to 200˚C by 6˚C/min and holding at 200˚C for 5 min, heating at 10˚C/min to 250˚C, and holding at 250˚C for 5 min. The instrument parameters were the following: split ratio = unsplit; inlet temperature = 250˚C; scanning mode = full scan; ion source temperature = 230˚C; quadrupole temperature = 180˚C; ion source = EI, 70 eV. The relative contents of each component were analyzed by the area normalization method.

## Determination of vitamins B1, B2, and E

The process used for the vitamin B (VB) assay was modified from the China national standard (GB 5009.84–2016). Corn juice (5 mL) was added to a centrifuge tube and dried at 60˚C under nitrogen air flow. Then, 4 mL of 2% formic acid was added and the mixture was ultrasonicated for 20 min. Next, 3 mL dichloromethane was added to mixture and vibrated for 2 min. The mixture was centrifuged at $5000 \times g$ for 10 min, and 1 mL supernatant was transferred to a C18 solid-phase extraction column and eluted with 2% ammonia in methanol. The eluent was dried by blowing nitrogen over the sample. The mobile phase consisted of solvent A (acetic acid/sodium acetate (1.8%/0.98%) solution) and solvent B (methyl alcohol). The elution conditions were as follows: 0–8 min, 90% solvent A; 8–15 min, solvent A was reduced from 90% to 65%; 15–25 min, solvent A was reduced from 65% to 50%; 25–25.1 min, solvent A was increased from 50% to 90%; 25.1–35 min, solvent A was held at 90%. All the gradients were linear and the flow rate was 1 mL/min. Finally, the dried compounds were redissolved in 1 mL water. Analysis was performed on an Agilent 1260 Infinity LC with a C18 (Shiseido, 4.6 mm × 250 mm × 5 μm) chromatographic column, DAD detector, 30˚C column temperature, 10 μL injection volume, and 270 nm test wavelength.

For vitamin E (VE) assay, the applied process was modified from the China national standard (GB/T 17812–2008). Corn juice (5 mL) was mixed with 100 mL ascorbic acid/ethyl alcohol solution (2 g ascorbic acid dissolved in 10 mL water and mixed with 90 mL ethyl alcohol) and 25 mL KOH solution (50 g KOH dissolved in 50 mL water). After 30 min, 50 mL petroleum ether was used to extract vitamin E. Water was then mixed with petroleum ether and then separated to remove the acid. The petroleum ether was dried successively by anhydrous sodium sulfate, rotary evaporation, and nitrogen flow. Finally, the dried compounds were dissolved in 10 mL methyl alcohol and filtered through a 0.22 μm membrane before analysis using a Thermo-U3000 HPLC with a C18 chromatographic column (Agilent, 4.6 mm × 150 mm × 5 μm). The instrument parameters were as follows: detector = DAD-FLD; column temperature = 20˚C; injection volume = 10 μL; flow rate = 1 mL/min; excitation wavelength = 294 nm; emission wavelength = 328 nm. The mobile phase was 98% methyl alcohol and 2% water.

## Total soluble sugar, free phenols, anthocyanins, and radical scavenging activity determination

Total soluble sugar content was determined using anthrone reagent with glucose as the standard [13]. Total free phenol content and anthocyanin content was determined using the Folin–Ciocalteau colorimetric method [21] and the pH differential method [22], respectively. Gallic acid and cyanidin 3-O-glucoside were used as the standards in phenols and anthocyanin determination, respectively.

DPPH scavenging activity was performed according to Mohapatra's method [16], and 1,2,3-trihydroxybenzene was used to determinate free radical scavenging activity by following Li's method [23].

## Anti-nutritional factors

Trypsin inhibition activity and phytic acid content were determined by following Dwivedi's method [24]. For trypsin inhibition activity, benzoyl-DL-arginine *p*-nitroanilide hydrochloride (BAPA) was used to detect trypsin activity by measuring absorbance at 410 nm using a UV-Vis spectrophotometer, and the trypsin inhibition activity was calculated by the difference between the sample and the control. For phytic acid content, Wade reagent (0.03% solution of $FeCl_3 \cdot 6H_2O$ containing 0.3% sulfosalicylic acid in water) was used, and absorbance was measured at 500 nm using a UV-Vis spectrophotometer. The final content was calculated based on the standard curve.

## Viscosity and stability

Fresh corn juice was diluted five-fold its volume, and half the quantity was then boiled. The raw and boiled homogenates were filtered through two layers of medical gauze (cotton fabrics with 1 mm aperture size), and the filtrate was used for viscosity and stability determination. The viscosity of the juice was determined using an Ubbelohde viscosity meter. The stability of the juice was determined from the change in turbidity after centrifugation for 10 min at 2000 × *g*. The transmittance of the corn juice was measured at 660 nm before (T0) and after (T10) centrifugation. The stability was calculated as T0/(T10-T0); the larger this ratio, the more stable the juice.

# Results and discussion

## Amino acid composition and total soluble sugar

The total amino acid and sugar contents are two important quality characteristics of sweet corn. At the milk stage, sweet corns have remarkably higher sugar and amino acid contents, particularly that of lysine, than normal field corns [2–5]. The amino acid composition and total soluble sugar contents of the raw and boiled sweet corn juices are presented in Table 1 and Fig 1, respectively. The total amino acid (TAA) for the two juices was comparable and the boiled juice contained more sugars. Generally, cooking processes cause a reduction of amino acids and soluble sugar in solid foods. This is particularly true for the boiling process because free amino acids and soluble sugars will leach into the surrounding water [13,17]. In the present study, the comparable TAA contents between boiled and raw juices implied that a short boiling time had no effect on the amino acid content. The proportion of total essential amino acids (TEAA) to TAA is an important indicator of protein quality. TEAA/TAA values of 39%, 26%, and 11% are considered adequate for infants, children, and adults, respectively [25]. The raw and boiled sweet corn juices, having TEAA/TAA values of 42.7–43.39%, were adequate for infants and comparable to wheat flour (44.7–47.3%) [26] and some fish (43.6–46.49%) [17], and were superior to sorghum grain (36.8–38.06%) [16].

The P-PER is another important indicator of protein quality, reflecting the potential efficiency of bioconversion of the tested protein into body weight. The experimentally determined PER usually ranges from 0 for a very poor protein to just over 4 for a high-quality protein [25]. Casein, a dairy protein and the standard reference for the PER, has a PER of 2.5 [25]. For the purple sweet corn used herein, the P-PER ranged from 2.34 to 2.84, comparable to those of wheat flour (2.19–2.46) [26] and some fish (2.29–2.60) [17], and superior to those of the sorghum grain (0.02–0.64) [16] and millet (1.32–1.66) [26].

The increased sugar content in the boiled juice may be due to the release of sugar from the sweet corn residue by thermal processing and/or hydrolysis of starch by amylase. It was consistent with prior reports, which stated that boiling promoted the leaching of sugars from

**Table 1. Amino acid composition in raw and boiled purple sweet corn juice.**

|  | Raw | | Boiled | |
|---|---|---|---|---|
|  | Mean+SD(mg/L) | %Total | Mean+SD(mg/L) | %Total |
| Asp | 612.53+78.06 | 5.58 | 571.07+19.75 | 5.48 |
| Glu | 1419.05+262.66 | 12.93 | 1319.70+75.91 | 12.66 |
| Cys | 1189.15+42.32 | 10.84 | 1359.27+100.37 | 13.04 |
| Ser | 449.79+82.13 | 4.10 | 399.61+10.25 | 3.83 |
| Gly | 399.73+42.27 | 3.64 | 382.47+10.40 | 3.67 |
| His[#] | 354.17+32.93 | 3.23 | 338.97+4.10 | 3.25 |
| Arg[#] | 574.89+176.68 | 5.24 | 538.52+86.44 | 5.17 |
| Thr[#] | 268.63+61.52 | 2.45 | 248.43+14.32 | 2.38 |
| Ala | 722.05+77.97 | 6.58 | 676.43+22.68 | 6.49 |
| Pro | 1418.52+58.18 | 12.93 | 1262.57+103.37 | 12.11 |
| Tyr[#] | 429.78+57.24 | 3.92 | 393.24+7.44 | 3.77 |
| Val[#] | 510.82+61.71 | 4.66 | 476.82+12.03 | 4.58 |
| Met[#] | 297.62+25.64 | 2.71 | 298.96+15.80 | 2.87 |
| Ile[#] | 406.21+33.65 | 3.70 | 372.52+14.18 | 3.57 |
| Leu[#] | 899.95+143.98 | 8.20 | 822.24+34.33 | 7.89 |
| Phe[#] | 477.30+64.83 | 4.35 | 443.53+10.00 | 4.26 |
| Lys[#] | 540.50+47.42 | 4.93 | 517.29+12.45 | 4.96 |
| **TEAA** | 4759.88 | 43.39 | 4450.52 | 42.70 |
| **TNEAA** | 6210.82 | 56.61 | 5971.12 | 57.30 |
| **TAA** | 10970.7 | 100 | 10421.64 | 100 |
|  | Raw | | Boiled | |
| **PER**(1) | 2.45 | | 2.34 | |
| **PER**(2) | 2.84 | | 2.72 | |

TEAA: total essential amino acids. TNEAA: total non-essential amino acids. TAA: total amino acids PER: protein efficiency ratio. Details were stated in Materials and methods. Superscript ([#]) represents essential and semi-essential amino acids for adults and infants.

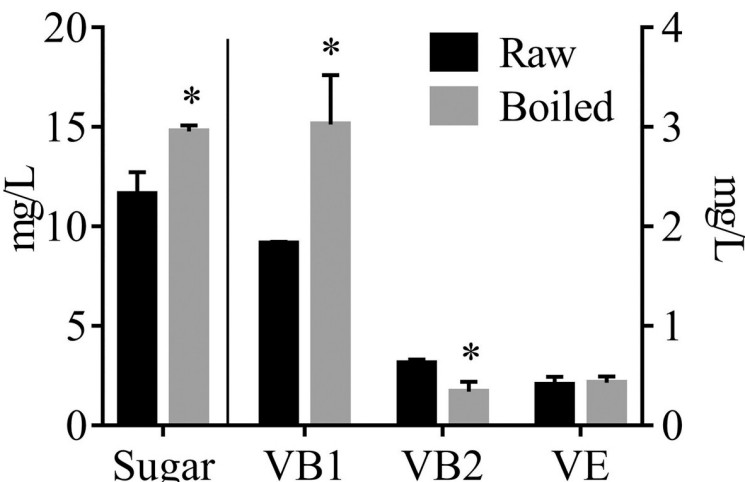

**Fig 1. The content of soluble sugar, vitamins B1 (VB1), B2 (VB2) and E (VE) in raw and boiled juices.** Asterisk (*) represents that the difference between raw and boiled juices was significant. * P<0.05.

broccoli into the water [13] and promoted the hydrolysis of starch into reducing sugars in sweet potatoes [27]. Another explanation is that large amounts of glycosyls were released into solution during the boiling-induced degradation of anthocyanins since purple corns contain high amounts of glycosylated anthocyanins [28].

## Vitamins B1, B2, and E

Vitamins are very important trophic factors in food. The contents of thiamin (vitamin B1), riboflavin (vitamin B2), vitamin B6, and vitamin E in yellow and white sweet corns are significantly lower than those in normal yellow and white field corns [5]. In this study, vitamins B1, B2, and E were determined in purple sweet corn juices. As shown in Fig 1, the contents of vitamins B1, B2, and E in the raw corn juice were 1.83, 0.62, and 0.41 mg/L, respectively. When the vitamin contents in the fresh corn kernels were calculated from the juice values, the contents of the vitamins B1, B2, and E were 5.22, 1.78, and 1.16 mg/kg, respectively. The vitamin B1 content in the purple sweet corn was higher than that reported in other sweet corns (white: 2 mg/kg, yellow: 1.55 mg/kg) and normal field corns (white: 3.85 mg/kg, yellow: 3.85 mg/kg) [5]. Moreover, the vitamin B2 and E contents in the purple sweet corn were higher than those reported in white (B2: 0.6 mg/kg, E: 0.7 mg/kg) and yellow (B2: 0.55 mg/kg, E: 0.7 mg/kg) sweet corns [5].

Vitamin B1 is water-soluble and thermally sensitive under moist conditions [18]. Therefore, there is a large loss of vitamin B1 upon boiling and steaming, but a trivial loss from brief microwave heating [29]. In contrast, vitamins B2 and E are more heat-stable than vitamin B1 under moist conditions, and only slightly affected by thermal processing. However, vitamin B2 is easily degraded by light, and vitamin E is susceptible to chemical modifications in the presence of reactive oxygen species [4,18,30]. The results from this study, however, were puzzling. After 5 min of boiling, the content of vitamin B1 increased from 1.83 to 3.02 mg/L, instead of decreasing as expected. In contrast, the content of vitamin B2, a heat-stable vitamin, decreased from 0.62 to 0.34 mg/L. Vitamin E content did not show any significant change. The increase of vitamin B1 content in the purple sweet corn juice after boiling was difficult to explain. Further research is needed to understand this phenomenon.

## Fatty acid composition

Fatty acid composition is an important indicator of food quality. Corn is reported to have high fatty acid content in the germ, with a fraction of unsaturated fatty acid (UFA) being as high as 80% [3,5]. Thus, corn oil is a high-quality edible oil. In addition, sweet corns have been reported to have higher fatty acid content at the milk stage than normal field corns [2,3].

As shown in Table 2, 35 different fatty acids were identified (fatty acids with a content less than 0.5 mg/L are showed as n). The amounts of the vast majority of fatty acids in the sweet corn juice decreased after boiling, with the largest reduction being ~57.6% for oleic acid. Among several other major components, the loss was 35.8% for linoleic acid, 55.3% for hexadecanoic acid, and 50.4% for stearic acid. However, the erucic acid content did not decrease after boiling. Considering the presence of aroma compounds (Table 3), it was initially speculated that boiling enhanced fatty acid loss by promoting their evaporation. This is because large amounts of oleic acid, linoleic acid, and hexadecanoic acid were identified in the aroma compounds of the boiling juice, but only a trace amount of that were found in the raw juice. However, it is not believed that evaporation of boiling water makes such a big difference in the content of long-chain fatty acids. Thus, the residues from corn juices were analyzed. As expected, most fatty acids showed significantly higher content in the boiled corn residue than in the raw corn residue, except for erucic acid (Table 2). The contents of oleic acid, linoleic

**Table 2. Fatty acid composition of raw and boiled purple sweet corn juice.**

| | Corn Juice | | | | Corn Residues | | | |
|---|---|---|---|---|---|---|---|---|
| | Raw | | Boiled | | Raw | | Boiled | |
| | Mean+SD (mg/L) | %Total | Mean+SD (mg/L) | %Total | Mean+SD (mg/kg Dry base) | %Total | Mean+SD (mg/kg Dry base) | %Total |
| C8.0 | 1.56+0.12 | 0.01 | n | n | 16.90+2.54 | 0.02 | 14.62+2.06 | 0.01 |
| C10.0 | n | n | n | n | 1.74+0.66 | 0.00 | 3.28+2.08 | 0.00 |
| C11.0 | n | n | n | n | n | n | n | n |
| C12.0 | 0.87+0.10 | 0.01 | 1.41+0.54 | 0.02 | 33.14+5.03 | 0.04 | 53.38+36.49 | 0.04 |
| C13.0 | n | n | n | n | n | n | n | n |
| C14.0 | 7.12+0.87 | 0.07 | 3.69+0.39 ** | 0.06 | 94.73+5.52 | 0.12 | 133.10+39.88 | 0.09 |
| C14.1 | n | n | n | n | n | n | n | n |
| C15.0 | 1.77+0.34 | 0.02 | 0.98+0.097 * | 0.02 | 20.10+0.74 | 0.03 | 27.42+3.93 * | 0.02 |
| C15.1 | 1.39+0.10 | 0.01 | 1.36+0.25 | 0.02 | n | n | n | n |
| C16.0 | 2609.80+455.17 | 24.11 | 1165.54+121.31 * | 19.88 | 18325.03+544.29 | 23.62 | 32344.69+6565.03 * | 21.30 |
| C16.1 | 13.87+3.25 | 0.13 | 7.13+2.01 | 0.12 | 182.97+8.12 | 0.24 | 326.19+88.97 * | 0.21 |
| C17.0 | 10.77+2.58 | 0.10 | 4.62+0.58 * | 0.08 | 100.61+1.17 | 0.13 | 172.85+28.79 * | 0.11 |
| C17.1 | n | n | n | n | 30.13+1.71 | 0.04 | 64.95+8.83 ** | 0.04 |
| C18.0 | 392.98+54.04 | 3.63 | 194.77+33.97 * | 3.32 | 1122.61+67.11 | 1.45 | 2196.95+355.13 * | 1.45 |
| C18.1N9C | 3329.52+779.29 | 30.76 | 1412.51+157.12 * | 24.09 | 16913.46+1168.66 | 21.80 | 33789.96+6922.85 * | 22.26 |
| C18.1N9T | 88.50+19.07 | 0.82 | 50.28+14.42 | 0.86 | 344.69+6.38 | 0.44 | 604.01+30.10 ** | 0.40 |
| C18.2N6C | 3476.72+866.66 | 32.12 | 2231.04+206.66 * | 38.05 | 38997.97+2098.46 | 50.26 | 79470.83+13970.82 * | 52.34 |
| C18.2N6T | n | n | n | n | 6.44+1.36 | 0.01 | 22.62+3.20 ** | 0.01 |
| C18.3N3 | 54.42+14.79 | 0.50 | 45.24+8.63 | 0.77 | 387.20+29.54 | 0.50 | 1267.48+152.41 ** | 0.83 |
| C18.3N6 | n | n | n | n | n | n | n | n |
| C20.0 | 69.05+13.25 | 0.64 | 27.93+5.81 * | 0.48 | 207.01+5.24 | 0.27 | 342.70+51.19 * | 0.23 |
| C20.1 | 29.48+4.09 | 0.27 | 13.08+1.50 ** | 0.22 | 111.49+7.46 | 0.14 | 192.06+34.98 * | 0.13 |
| C20.2 | 2.85+0.48 | 0.03 | 1.57+0.23 * | 0.03 | 11.71+0.56 | 0.02 | 22.33+3.88 * | 0.01 |
| C20.3N3 | n | n | n | n | n | n | n | n |
| C20.3N6 | 1.50+0.06 | 0.01 | 1.22+0.11 * | 0.02 | n | n | n | n |
| C20.4N6 | n | n | n | n | n | n | n | n |
| C20.5N3 | 1.62+0.37 | 0.01 | 0.98+0.16 | 0.02 | n | n | n | n |
| C21.0 | 1.52+0.25 | 0.01 | 1.02+0.16 | 0.02 | 3.56+0.49 | 0.00 | 5.01+0.83 | 0.00 |
| C22.0 | 34.34+5.88 | 0.32 | 19.78+3.13 * | 0.34 | 119.98+17.25 | 0.15 | 131.66+36.12 | 0.09 |
| C22.1N9 | 619.55+81.37 | 5.72 | 650.80+84.61 | 11.10 | 493.21+26.05 | 0.64 | 554.79+58.79 | 0.37 |
| C22.2 | 13.20+2.31 | 0.12 | 14.21+2.64 | 0.24 | 11.55+1.00 | 0.01 | 12.73+2.06 | 0.01 |
| C22.6N3 | n | n | n | n | n | n | n | n |
| C23.0 | 2.62+0.64 | 0.02 | 1.40+0.36 | 0.02 | 4.88+0.03 | 0.01 | 6.97+1.17 * | 0.00 |
| C24.0 | 47.26+10.61 | 0.44 | 20.08+2.12 * | 0.34 | 44.05+4.36 | 0.06 | 60.92+16.60 | 0.04 |
| C24.1 | 12.40+5.64 | 0.11 | 9.01+0.93 | 0.15 | n | n | n | n |
| **TSFA** | 3179.66 | 29.37 | 1441.22 | 24.58 | 20094.35 | 25.90 | 35493.55 | 23.38 |
| **TMUFA** | 4094.71 | 37.83 | 2144.17 | 36.57 | 18075.96 | 23.30 | 35531.96 | 23.40 |
| **TPUFA** | 3550.31 | 32.80 | 2278.48 | 38.85 | 39414.88 | 50.80 | 80796.00 | 53.22 |
| **TFA** | 10824.68 | | 5863.87 | | 77585.19 | | 151821.51 | |
| **Total FA of dry kernels** | 74.47 mg/g | | | | | | | |

TSFA: total saturated fatty acid; TMUFA: total monounsaturated fatty acid; TPUFA: total polyunsaturated fatty acid; TFA: total fatty acid. Asterisk (*) represents difference between raw and boiled was significant.

* P<0.05

** P<0.01.

**Table 3. Volatile compounds of raw and boiling purple sweet corn juice.**

| | Raw | | | Boiled | | |
|---|---|---|---|---|---|---|
| | 1 | 2 | 3 | 1 | 2 | 3 |
| **Alcohols, Aldehydes and Ketones** | | | | | | |
| 1,3-Propanediol, 2-(hydroxymethyl)-2-nitro- | n | n | n | n | 0.21 | n |
| 1,4-Pentadien-3-ol | n | n | n | 0.49 | n | n |
| 1,6-Heptadien-4-ol | n | n | n | n | n | 0.42 |
| 1-Heptanol | 0.78 | 0.88 | 0.66 | n | n | 0.28 |
| 1-Hexanol | 2.59 | 2 | 2.62 | 0.54 | n | 0.39 |
| 1H-Inden-1-one, octahydro-7a-hydroxy- | n | n | 1.55 | n | n | n |
| 1-Octadecanol | n | 0.09 | n | n | n | n |
| 1-Octanol | 0.73 | 0.57 | 0.49 | 0.29 | n | 0.31 |
| 1-Octen-3-ol | 5.15 | 4.11 | 4.54 | n | n | 0.72 |
| 1-Pentanol | 1.68 | n | 1.89 | n | n | n |
| 2,3-Octanedione | n | n | n | n | 0.23 | n |
| 2,4-Nonadienal, (E,E)- | n | 0.15 | 0.21 | n | n | n |
| 2-Butenal | 0.47 | n | n | n | n | n |
| 2-Butenal, 2-methyl- | n | 0.95 | 1.2 | 1.56 | n | 1.53 |
| 2-Cyclohexen-1-one, 3-methyl- | 0.51 | n | n | n | n | n |
| 2-Cyclopenten-1-one | n | n | 0.35 | n | n | n |
| 2-Ethylidenecyclohexanone | n | n | 24.03 | n | 1.48 | n |
| 2-Furancarboxaldehyde, 5-(hydroxymethyl)- | n | n | n | n | 0.25 | n |
| 2-Furanmethanol, tetrahydro- | n | n | n | n | 0.21 | n |
| 2-Heptanone | 2.14 | 2.5 | 3.35 | n | n | n |
| 2-Hepten-4-ol | n | 1.59 | n | n | n | n |
| 2-Heptenal, (Z)- | n | n | n | 0.39 | 0.3 | 0.39 |
| 2-Nonenal, (E)- | 3.55 | 2.42 | 1.09 | 1.32 | 0.69 | 0.83 |
| 2-Octanone | n | 3.81 | n | 0.67 | 1.02 | 1.06 |
| 2-Octen-1-ol, (E)- | n | n | n | n | n | 0.24 |
| 2-Octenal, (E)- | 1.63 | 1.47 | 1.5 | 0.65 | 0.38 | 0.5 |
| 2-Pentadecanone | n | n | n | n | 0.23 | n |
| 3,5-Octadien-2-ol | n | n | 0.68 | n | n | n |
| 3,5-Octadien-2-one, (E,E)- | 1.11 | 1.02 | 1.05 | n | n | n |
| 3-Nonen-1-ol, (Z)- | n | 0.29 | 0.14 | 0.6 | 0.31 | 0.52 |
| 3-Octanol, 2-methyl- | n | 0.31 | n | n | n | n |
| 3-Octen-2-one | 0.3 | 0.48 | n | n | n | n |
| 4-Methyl-2,5-dimethoxybenzaldehyde | n | n | n | 0.57 | 0.82 | n |
| 5-Methyl-5-hexen-3-yn-2-ol | n | 0.18 | n | n | n | n |
| 6-Dodecanol | n | n | n | n | n | 0.22 |
| 9,12-Octadecadien-1-ol, (Z,Z)- | n | n | n | n | n | 0.59 |
| 9,17-Octadecadienal, (Z)- | n | n | n | n | 0.18 | n |
| Cis-3-methylpent-3-ene-5-ol | 2.25 | n | n | n | n | n |
| Cyclohexanone | 1.36 | n | n | n | n | n |
| Cyclohexanone, 2-butyl- | n | n | n | 0.73 | n | 0.7 |
| Cyclohexanone, 2-methyl-5-(1-methylethyl)- | n | n | n | n | 0.59 | n |
| Decanal | 0.26 | 0.11 | n | 0.33 | n | n |
| Ethanone, 1-(2-hydroxy-4,6-dimethoxyphenyl)- | n | n | n | 0.47 | n | n |
| Ethanone, 1-(3-methoxyphenyl)- | n | n | n | 0.97 | n | n |
| Heptanal | 11.49 | 9.87 | 9.88 | 1.47 | 1.01 | 1.13 |

(*Continued*)

**Table 3.** (*Continued*)

| | Raw | | | Boiled | | |
|---|---|---|---|---|---|---|
| | **1** | **2** | **3** | **1** | **2** | **3** |
| Heptanol | n | n | n | n | 0.45 | n |
| Hexanal | 18.92 | 23.19 | 21.38 | 0.59 | 0.57 | 0.51 |
| Nonanal | 1.76 | n | 1.62 | 1.11 | 0.53 | 0.81 |
| Octanal | n | 1.05 | n | n | 0.36 | n |
| Silanediol, dimethyl- | 4.74 | n | n | n | n | 0.96 |
| Tridecanal | n | n | n | 0.27 | n | n |
| Vanillin | n | n | n | n | 0.2 | n |
| **Aliphatic Acids and Lactones** | | | | | | |
| 1,2-Benzenedicarboxylic acid, butyl 2-methylpropyl ester | n | n | n | 0.3 | n | n |
| 1,2-Benzenedicarboxylic acid, butyl octyl ester | 0.44 | n | n | n | n | n |
| 11,14-Eicosadienoic acid, methyl ester | 0.19 | n | n | n | n | n |
| 2,4-Hexadienoic acid, ethyl ester | n | n | 0.12 | n | n | n |
| 2-Propenoic acid, 3-(dimethylamino)-, methyl ester | n | 1.87 | n | n | n | n |
| 3-Nonenoic acid | n | n | n | n | 0.29 | 0.26 |
| 6-Octadecenoic acid, (Z)- | n | n | n | n | n | 23.09 |
| 9,12-Octadecadienoic acid(Z,Z)- | n | n | n | n | 17.94 | 15.19 |
| 9-Octadecenoic acid, (E)- | n | n | n | n | 16.34 | n |
| Acetic acid | 2.94 | 3.38 | n | 0.36 | n | n |
| Benzoic acid, 2,4-bis[(trimethylsilyl)oxy]-, trimethylsilyl ester | n | n | 0.13 | n | n | n |
| Butyric acid, 1-propylpentyl ester | n | n | 1.87 | 0.54 | n | n |
| Carbamic acid, methyl ester | n | n | n | n | 3.89 | n |
| Dodecanoic acid | n | n | n | 1.64 | 0.48 | 0.4 |
| Egtazic Acid | n | n | n | n | 0.21 | n |
| Formic acid, pentyl ester | n | 1.21 | n | n | n | n |
| Heptanoic acid | n | 0.51 | n | 0.35 | 0.23 | 0.29 |
| Hexanoic acid | n | n | 1.71 | 0.89 | n | n |
| n-Hexadecanoic acid | 1.63 | 0.33 | 0.27 | 41.64 | 40.06 | 36.94 |
| Nonanoic acid | n | n | n | 0.84 | 1.4 | 0.59 |
| Octadecanoic acid | n | n | n | 1.1 | n | 1.24 |
| Octanoic Acid | 0.96 | 0.68 | 0.5 | 0.85 | 0.36 | 0.47 |
| Oleic Acid | 0.59 | n | n | 16.08 | n | n |
| Tetradecanoic acid | n | n | n | n | 0.23 | 0.25 |
| **Aliphatic hydrocarbon** | | | | | | |
| 1,3-Hexadiene, 3-ethyl-2-methyl- | 19.2 | 24.75 | n | n | n | 2.07 |
| 1-Nonen-3-ol | n | n | n | n | 0.58 | n |
| 1-Pentene, 3-ethyl-3-methyl- | n | 1.45 | n | n | n | n |
| 2,4-Hexadiene, 2,5-dimethyl- | n | 0.09 | n | n | n | n |
| 2-Pentene, 3,4-dimethyl-,(Z)- | n | n | 0.36 | n | n | n |
| 2-Pentene, 4,4-dimethyl-,(E)- | 0.38 | n | n | n | n | n |
| 2-Pentene, 4-methyl-,(Z)- | n | 0.1 | n | n | n | n |
| 5,5-Dimethyl-1,3-hexadiene | 0.25 | n | n | n | 0.2 | n |
| 7-Hexadecyne | n | n | n | 10.56 | n | n |
| 8-Oxabicyclo[5.1.0]octane | n | 0.15 | n | n | n | n |
| Cyclodecene, (E)- | n | n | n | 0.4 | n | n |
| Cyclooctane | n | n | 1.15 | n | n | n |

(*Continued*)

**Table 3.** (Continued)

| | Raw | | | Boiled | | |
|---|---|---|---|---|---|---|
| | 1 | 2 | 3 | 1 | 2 | 3 |
| **Aliphatic hydrocarbon** | | | | | | |
| Cyclooctene, 3-methyl- | 0.46 | n | n | n | n | n |
| Cyclopentane, 1,1,3-trimethyl- | n | 1.8 | n | n | n | n |
| Cyclopentane, 1-ethyl-2-methyl-, cis- | 1.58 | n | n | n | n | n |
| Cyclopentene, 1-butyl- | n | n | n | 2.67 | n | n |
| Heneicosane | n | n | n | n | 0.48 | n |
| Heptadecane | n | n | n | n | n | 0.21 |
| Heptadecane, 2,6,10,15-tetramethyl- | n | n | n | 0.43 | n | n |
| Hexadecane | n | n | n | 0.37 | n | 0.21 |
| Spiro[4.4]nonane-1,6-dione | n | 0.35 | n | n | n | n |
| trans-1,4-Hexadiene | 0.26 | n | n | n | n | n |
| Triallylvinylsilane | n | n | 0.1 | n | n | n |
| **Nitrogen, Furans, Pyrans, and Aromatic Compounds** | | | | | | |
| 1,2-Benzenediol, 3,5-bis(1,1-dimethylethyl)- | n | n | 0.34 | n | n | n |
| 1,2-Bis(trimethylsilyl)benzene | 2.19 | 0.59 | n | n | n | n |
| 1,3-Propanediamine, N-methyl- | n | n | 0.14 | n | n | n |
| 1-Butanamine, N-butylidene- | n | n | n | n | 2.06 | n |
| 1H-Indole, 1-methyl-2-phenyl- | n | 1.98 | n | 3.34 | n | n |
| 1-Methyl-4-[nitromethyl]-4-piperidinol | n | n | 0.09 | n | n | n |
| 2(3H)-Furanone, dihydro-5-pentyl- | 0.88 | 1.09 | 0.98 | 1.91 | 2.06 | 0.86 |
| 2,3,5,6-Tetrafluoroanisole | n | n | n | n | n | 0.66 |
| 2-Amino-6-methylbenzoic acid | n | n | n | n | n | 0.91 |
| 2-Methoxy-4-vinylphenol | n | n | n | n | 1.28 | 1.3 |
| 2-Nonenenitrile | n | 0.12 | n | n | 0.21 | 0.23 |
| 2-Propanamine | n | n | 8.2 | n | n | n |
| 3,3'-Iminobispropylamine | n | n | 0.06 | n | n | n |
| 4-Amino-6-hydroxypyrimidine | 0.81 | n | 0.95 | n | n | n |
| 5H-Naphtho[2,3-b]carbazole | n | n | 0.11 | n | n | n |
| 5-Methyl-2-phenylindolizine | n | n | 0.38 | n | n | n |
| 6H-Pyrazolo[1,2-a][1,2,4,5]tetrazine, hexahydro-2,3-dimethyl- | n | n | 0.1 | n | n | n |
| 7H-Dibenzo[b,g]carbazole, 7-methyl- | n | n | n | n | n | 0.32 |
| Auramine o | 0.19 | 0.15 | n | n | n | n |
| Benzaldehyde | n | n | n | n | 0.2 | n |
| Benzaldehyde, 2,5-bis[(trimethylsilyl)oxy]- | 0.47 | n | n | n | n | n |
| Benzene, 1-methoxy-4-(1-propenyl)- | 0.19 | n | n | 0.77 | 0.25 | n |
| Benzeneacetaldehyde | n | 0.1 | n | n | n | n |
| Benzofuran, 2,3-dihydro- | n | n | n | n | 0.19 | n |
| Furan, 2-(dichloromethyl)-tetrahydro- | n | 2.14 | n | n | n | n |
| Furan, 2-pentyl- | 3.62 | n | 4.15 | n | n | n |
| Hexahydropyridine, 1-methyl-4-[4,5-dihydroxyphenyl]- | n | n | 0.11 | n | n | n |
| Indolizine | n | n | n | 0.34 | n | n |
| Phenol, 2-(1-methylpropyl)- | n | n | n | n | n | 0.36 |
| Phenol, 2,4-bis(1,1-dimethylethyl)- | n | n | n | 1.26 | 0.78 | 1.39 |
| Phenol, 2,5-bis(1,1-dimethylethyl)- | n | n | 0.08 | n | n | n |
| Phenol, 4-(1-methylpropyl)- | n | n | n | 0.35 | 0.46 | n |
| Propanenitrile, 3-(methylamino)- | n | n | 0.21 | n | n | n |

(*Continued*)

**Table 3.** (Continued)

| | Raw | | | Boiled | | |
|---|---|---|---|---|---|---|
| | 1 | 2 | 3 | 1 | 2 | 3 |
| **Nitrogen, Furans, Pyrans, and Aromatic Compounds** | | | | | | |
| Pyridine, 4-methoxy- | n | 0.09 | n | n | n | n |
| Silane,[[4-[1,2-bis[(trimethylsilyl)oxy]ethyl]-1,2-phenylene]bis(oxy)]bis[trimethyl- | 0.27 | n | n | n | n | n |
| s-Triazole, 3-acetamido- | 1.07 | n | n | n | n | n |
| trans-4-Dimethylamino-4'-methoxychalcone | n | n | n | n | n | 0.28 |

The compounds with gray background had been reported in previous studies.

acid, hexadecanoic acid, and stearic acid increased by 99.78%, 103.78%, 76.51%, and 95.70%, respectively. Based on these results, it is proposed that boiling process greatly increases the retention of fatty acids in corn residues, potentially by forming covalent linkages or by binding with cell wall components, as fatty acids are known to react with cellulose via esterification [31].

On a dry weight basis, the total fatty acid content in this purple sweet corn was ~7%, which is lower than that in high-oil corn (~10%) and higher than that in most normal field corns (~5%) [3,32,33]. However, the content (~24%) of hexadecanoic acid, the main component of saturated fatty acids, in the purple sweet corn was notably higher than those reported for white and yellow corn (less than 10%) [3,32]. Thus, the proportion of UFA in the purple sweet corn was remarkably lower than that in normal field corns. There is a potential possibility that the biosynthesis of anthocyanin may have adverse effects on the accumulation of UFA, and further research is needed for verification.

## Aroma compounds

The relative amounts of the aroma compounds in three biological replicates of raw and boiled juices are individually shown in Table 3. A total of 136 compounds, a noticeably larger quantity than the 43–87 compounds reported in previous papers, were identified from six samples. Among these compounds, only 46 had been reported previously in sweet corns, popcorns, corn tortillas, and related products [34–36]. The identification of a large number of new compounds in this study may be attributed to advances in identification techniques and sufficient biological replicates. The presence of most compounds varied greatly between three biological replicates used in this study, and most of the new compounds were only sporadically identified in the three replicates, suggesting they occurred in trace amounts. In addition, 20 compounds, of which 17 had been previously reported, could be consistently identified in all three replicates of the raw or boiled juice. The major components identified in the purple sweet corn were hexanal, 1-octen-3-ol, 1-hexanol, 2-heptanone, and (E)-2-nonenal, which is consistent with the major components previously identified in corns. Moreover, some previously reported major compounds of corn, such as 1-hydroxy-2-propanone, 3-hydroxy-2-butanone, 2,3-butanediol, and acetaldehyde, were not identified in this study [34–36].

Five of the newly identified compounds, 1-heptanol, 2-methyl-2-butenal, (Z)-3-nonen-1-ol, 3-ethyl-2-methyl-1,3-hexadiene, and 2,4-bis(1,1-dimethylethyl)phenol, were consistently detected in the biological replicates. In particular, 3-ethyl-2-methyl-1,3-hexadiene accounted for 20% of the volatile compounds and was approximately equal in amount to the major component hexanal, which has been consistently reported to occur [34–36]. Furthermore, in a previous study, heptanal accounted for a very small proportion of the aroma compounds [34–36],

whereas in the present work it was found to be a major component (~10%) of the aroma compounds. Collectively, there is a big difference in the aroma compound composition for the purple sweet corn investigated in the present work and those of previously reported corns, which may be the cause of different aromas between normal field corn and sweet corn [34–36].

Boiling promotes the release of aroma compounds from corn juice, thus enhancing properties related to desirable odors. In addition, boiling changes the composition of the volatile compounds through Maillard reactions [37]. Consequently, forty-nine compounds were identified exclusively in the boiling corn juice, such as (Z)-2-heptenal, dodecanoic acid, nonanoic acid and 2,4-bis(1,1-dimethylethyl)-phenol. Fifty-four compounds, such as 1-pentanol, 2-heptanone, (E,E)-3,5-octadien-2-one, and 2-pentyl-furan, were identified exclusively from the raw corn juice. Thus, a difference in aroma due to the boiling of purple sweet corn juice can be expected.

## Phenols, anthocyanins, and antioxidant activity

The total free phenols content and antioxidant activity of the purple sweet corn juice decreased after boiling (Table 4; Frozen juice). This conflicts with Dewanto's report that thermal processing in sealed cans at 100, 115, and 121˚C for 25 min remarkably elevated the total free phenols content and antioxidant activity of yellow sweet corn kernels [11]. In the cited work, severe heat treatment was believed to promote the release of bound phenols via an autohydrolysis reaction. However, a shorter heating time and lower temperature (95˚C for 5 min) was applied in the present work. The investigated process conditions may have been insufficient for the release of bound phenols from corn residues. The change in anthocyanins content also differed from what is reported in previous literature, as anthocyanins are generally heat-sensitive and readily degrade during thermal processing [22,38]. However, the present work found that boiling for 5 min increased the total anthocyanin content in purple sweet corn juice (Table 4; Frozen juice).

Enzymatic degradation of anthocyanins is ubiquitous in tissue homogenate [38]. Heat blanching of a purple corn cob for 4 min and 10 min reduced peroxidase activity by 99% and 100%, respectively, and hence, elevated the stability of the anthocyanins [38]. Thus, we speculated that a short period of heat blanching may inactivate enzymes, such as anthocyanase, polyphenol oxidase, and peroxidase, and consequently reduce the degradation of anthocyanins during storage (approximately 20 days at -80˚C). Therefore, fresh (unrefrigerated) juice was used to determine the total free phenols content, anthocyanins content, and antioxidant activity. As shown in Table 4 (Fresh juice), the phenols and anthocyanins content decreased by 27.3% and 21.0%, respectively, after boiling when using fresh juice. Accordingly, the antioxidant activity of boiled fresh juice decreased by 39.5% and 22.0%, as determined by the DPPH and trihydroxybenzene methods, respectively. However, in freeze-preserved juice, the phenol content decreased by 19.2% and the anthocyanin content increased by 26.3%. Consequently,

**Table 4. Free phenols, anthocyanins and antioxidant activity of raw and boiled purple sweet corn juice.**

| | Frozen juice (Mean+SD) | | Fresh juice (Mean+SD) | |
|---|---|---|---|---|
| | Raw | Boiled | Raw | Boiled |
| Phenols | 0.292+0.011[Bb] | 0.236+0.011[Cc] | 0.341+0.010[Aa] | 0.248+0.013[Cc] |
| Anthocyanins | 0.038+0.005[Bc] | 0.048+0.007[ABbc] | 0.062+0.008[Aa] | 0.049+0.005[ABb] |
| Antioxidant-DPPH | 33.697+1.679[Ab] | 21.655+0.834[Bc] | 38.886+1.040[Aa] | 23.527+3.099[Bc] |
| Antioxidant-Trihydroxybenzene | 28.348+0.935[Ab] | 23.437+0.765[Bc] | 31.847+0.966[Aa] | 24.854+0.843[Bc] |

Phenols and anthocyanins content were expressed as mg gallic acid equivalent per mL of the juice and mg cyanidin 3-O-glucoside equivalent per mL of the juice, respectively. Antioxidant activity was expressed as free radical scavenging percentage. Superscripts on numerical values are notes of significance test of different treatments, lowercase letters: P < 0.05; uppercase letters: P < 0.01.

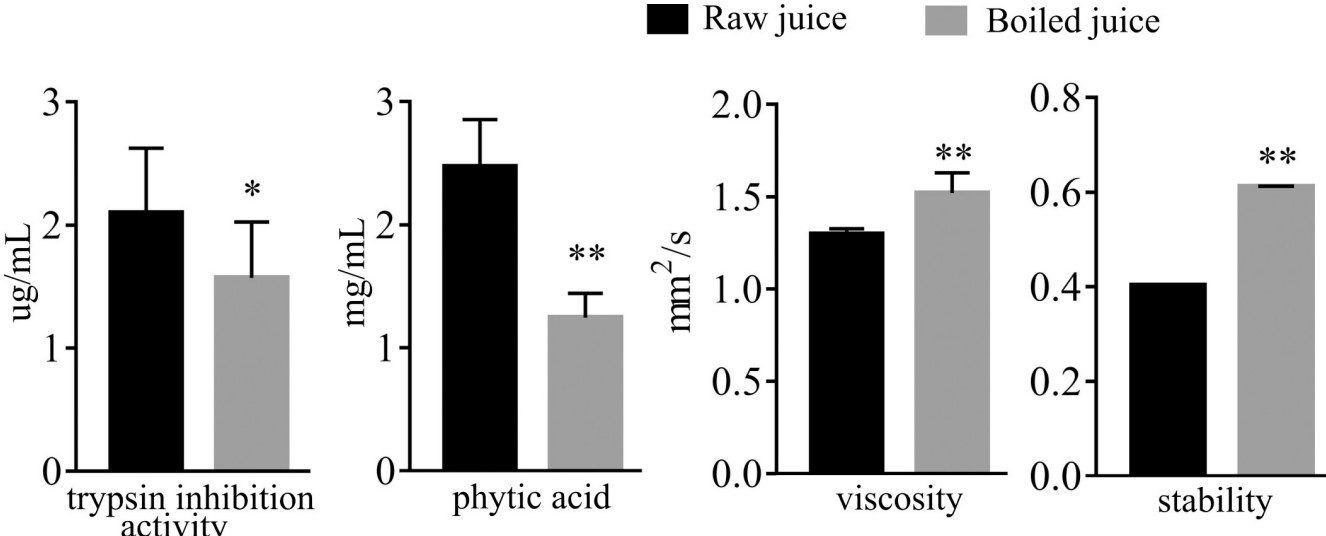

**Fig 2. The content of two anti-nutritional factors, viscosity and stability of juices.** Asterisk (*) indicates significant difference between raw and boiled samples. * P<0.05, ** P<0.01.

the antioxidant capacity decreased after boiling by 35.7% and 17.3% as determined by the DPPH and trihydroxybenzene methods, respectively. In addition to that of anthocyanins, the degradation of phenols during refrigerated storage is also reduced by boiling process. However, boiling itself has an adverse impact on phenols, anthocyanins, and antioxidant activity in corn juice. This confirmed that the degradation of anthocyanins and phenols happened during refrigeration in the raw juice but not the boiled juice, and a boiling process is necessary for a purple corn juice which needs long-term refrigeration (Table 4 and S1 Table).

## Anti-nutritional factors

Trypsin inhibitors and phytic acid are two primary anti-nutritional factors in corn [39]. They have inhibitory effects on digestive enzymes and can retard mineral absorption, thus limiting the bioavailability of nutrients [16]. The trypsin inhibition activity and phytic acid content in the raw and boiled juices are presented in Fig 2. Trypsin inhibition activity and phytic acid content were decreased by 50% and 55%, respectively, for boiled corn juice. These results are consistent with previous reports that thermal processing effectively reduces anti-nutritional factors [16]. Therefore, the nutrients in boiled corn juice are better absorbed by the body.

## Viscosity and stability

Thermal processing at 95°C for 5 min promotes the gelatinization of starch in corn juice [40]. Gelatinized starch will change the physical properties of the juice, such as viscosity and stability. Therefore, the viscosity and stability of the purple sweet corn juice were tested. As shown in Fig 2, boiling process slightly elevated both viscosity (from 1.29 to 1.52 mm$^2$/s) and stability (from 0.40 to 0.61) of the purple sweet corn juice. These physical changes are beneficial because the appearance of the beverage is more homogenous and stable.

## Conclusions

This study offers a comprehensive understanding of the advantages and disadvantages of boiling purple sweet corn juice. Boiling improved the stability of the juice, and potentially favored

enhanced odor by releasing aroma compounds from the juice into the air. Moreover, several aroma compounds unreported in previous studies were identified in the purple sweet corn, such as 1-heptanol, 2-methyl-2-butenal, (*Z*)-3-nonen-1-ol, 3-ethyl-2-methyl-1,3-hexadiene, and 2,4-bis(1,1-dimethylethyl)-phenol. The sugar content was elevated, and anti-nutritional factors were to a large extent removed by boiling. Unexpectedly, the fatty acid content in the boiled juice was remarkably lower than that in the raw juice because boiling process greatly increased the retention of fatty acids in the corn residues. In purple sweet corn juice, the total free phenols and anthocyanins contents, along with the antioxidant activity, were reduced after boiling. Phenols and anthocyanins in the raw juice were degraded during refrigerated storage. However, heat blanching for a short time significantly suppressed the degradation process and had no apparent effect on the amino acid composition and vitamin E content of the corn juice.

## Supporting information

**S1 Table. Two-way ANOVA of free phenols, anthocyanins and antioxidant activity.** (DOCX)

## Acknowledgments

We thank Dr. Xuejun Hua, Ioannis Dogaris, Leonidas Matsakas, and two anonymous reviewers for their suggestions and language polishing.

## Author Contributions

**Conceptualization:** Xuanjun Feng, Wei Guo.

**Data curation:** Liteng Pan, Xuemei Zhang.

**Funding acquisition:** Yanli Lu.

**Investigation:** Liteng Pan.

**Methodology:** Xuanjun Feng.

**Resources:** Qingjun Wang, Zhengqiao Liao, Xianqiu Wang, Erliang Hu, Jingwei Li.

**Supervision:** Wei Guo.

**Validation:** Xuemei Zhang, Jie Xu, Fengkai Wu.

**Writing – original draft:** Xuanjun Feng.

**Writing – review & editing:** Xuanjun Feng, Yanli Lu.

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
