## [Decision Letter · Decision Letter 0]

31 Mar 2020

PONE-D-20-06814

Nutritional and physicochemical characteristics of purple sweet corn juice before and after boiling

PLOS ONE

Dear Miss Lu,

Thank you for submitting your manuscript to PLOS ONE. After careful consideration, we feel that it has merit but does not fully meet PLOS ONE’s publication criteria as it currently stands. Therefore, we invite you to submit a revised version of the manuscript that addresses the points raised during the review process.

The experts that reviewed your article have raised some questions and made several suggestions on how to improve your manuscript. Please go through their comments and make the necessery amendments. 

We would appreciate receiving your revised manuscript by May 15 2020 11:59PM. To enhance the reproducibility of your results, we recommend that if applicable you deposit your laboratory protocols in protocols.io, where a protocol can be assigned its own identifier (DOI) such that it can be cited independently in the future. For instructions see: http://journals.plos.org/plosone/s/submission-guidelines#loc-laboratory-protocols

We look forward to receiving your revised manuscript.

Kind regards,

Leonidas Matsakas

Academic Editor

PLOS ONE

Journal Requirements:

Reviewers' comments:

Reviewer's Responses to Questions

**Comments to the Author**

1. Is the manuscript technically sound, and do the data support the conclusions?

Reviewer #1: Partly

Reviewer #2: Yes

Reviewer #3: Yes

2. Has the statistical analysis been performed appropriately and rigorously? 

Reviewer #1: I Don't Know

Reviewer #2: Yes

Reviewer #3: Yes

3. Have the authors made all data underlying the findings in their manuscript fully available?

Reviewer #1: Yes

Reviewer #2: Yes

Reviewer #3: No

4. Is the manuscript presented in an intelligible fashion and written in standard English?

Reviewer #1: Yes

Reviewer #2: Yes

Reviewer #3: Yes

5. Review Comments to the Author

Reviewer #1: Comment 1: Please upload all underlying data as attachments to the PLOS ONE submission site.

Else, please addresse the comments below.

L24-44: The abstract should not include methodological details. Please rephrase/remove. See author guidelines.

L25-28: Text: In order to contribute to the increased knowledge available for consumers on sweet corn juice, the nutritional and physicochemical characteristics of raw and boiled purple sweet juice was herein investigated.

L33-34: Refrigerated samples? Is this the freezed or raw samples? Both of those samples reach similar levels on anthocyanin after boiling. Please rephrase.

L40: Text: The boiling process had no … However, the content of fatty acids were …

L42-44: Text: These results provide detailed information relevant for further development of sweet corn juice …

L77: Text: Boiling and roasting has been found to…

L92: Is heat-blanching investigated in the present work? If not remove/rephrase.

L100-109: CAS numbers for the products and their purity/quality is missing.

L117: What type of juice extractor? Can I expect the same homogenization if I use my extractor at 10 000 rpm for 1 min?

L120: Medical gauze? Details? Pore-size? Material? This will affect the degree of filtration.

L126-129: Level of significance used (alpha value(s))? As mentioned in the guidelines, this should be specified.

L132: How is degradation and racemization of amino-acids accounted for in the applied method? Such effects will affect the estimated nutritional values of the juice.

L132-134: 1 mL of water was added to the dried/evaporated 1 mL aliquot, then added 1 mL of water before a second evaporation to dryness? This should be written clearer.

L134: What is the hydrolysate? The original liquid from which 1 mL of supernatant was withdrawn? Or the aliquot eventually evaporated? I guess both are hydrolysates and the difference/similarity should be stated clearer.

L140: What type of filter paper? Technical details?

L154: Equations should be numbered, referred to in the text, and described in detail.

L164: Ether-petroleum ether mixture? Diethyl ether mixed with petroleum ether? Which ratio?

L171: Added to the extraction and staying at 85 °C? Added to the separated and dried organic phase and kept at 85 °C for 30 min? Please clarify.

L172: Text: … and kept at 85 …

L177: Membrane material?

L181: Text: Heating to 200 °C…

L194: Text: Heating to …

L196: Text: Split ratio = unsplit; inlet temperature = 250 °C; ….

L219: Text: … to remove the acid.

L224: Text: Detector = DAD-FLD; column temperature = …

L239-240: Please give a short summary of the applied method.

L243: Technical details for medical gauze is missing.

L244: What are the units for the obtained viscosity?

L247: Why was absorbance used and why 660 nm? Which component in the juice absorbs light at this wavelength and why is this related to the stability of the juice?

L271: Table 1: Please clarify (with details, or correct) how there is a statistical difference (significance level of 0.05) between for example Raw vs. Boiled samples and their Methionine content considering the concentration in the juice is given as 297.62 ± 25.64 g/L vs. 298.96 ± 15.80 g/L, respectively. The same holds for Thr, Lys, Phe, Leu, …

L280: Text: Casein, a dairy protein…

L596: Figure 1 contain abbreviations not described in the text (VB1, VB2 and VE) and Figure text is missing for both Figure 1 and 2. Caption for Figure 2 is on L447, while caption for Figure 1 is missing. Please follow instructions given in the guidelines.

L301-303: I guess the values obtained for sweet and normal field corns are obtained from reference [5]? Anyways, please include the reference subsequent the states values, or at the end of L303.

L308: Text: … loss of vitamin B1 upon boiling and steaming…

L310: Text: … and only slightly affected by thermal processing.

L311: Text: … is easily degraded by light…

L312: Text: … is susceptible for chemical modifications in …

L321: Text: … in the germ, with a fraction of... being as high as…

L315: Text: … 35 different fatty acids were identified (fatty acids with a content less than 0.5 mg/L are not included)…

L330-333: Text: Considering the presence of aromatic compounds (Table 3), it was initially speculated that boiling enhanced fatty acid volatility, promoting their evaporation. This because …

L335-337: Please avoid personal pronouns. Text: However, it is believed that… What is the volatility of long chained fatty acids? Boiling points or vapour pressure for pure components? Text: Thus, the residues from … were also analysed.

L341: Avoid ‘we’.

L341-344: Very interesting!

L345: Text: On a dry weight basis, …

L358: Three biological replicates of what? Of raw and boiled purple sweet corn juice? Clarify.

L364: Text: The presence of most compounds varied…

L366-367: Text: …, suggesting they occurred in trace amounts.

L369: Text: … in the purple sweet …

L375: Table 3: Units are missing. Which internal standard was used during GC-MS for fatty acids and volatile components? Please include this information in the appropriate sections within the Materials and methods.

L382: Text: … which has been reported to occur persistently [References].

L384: Text: … whereas in the present work it was found to be a major component (~10%) within the aromatic compounds.

L386: References relating to the statement regarding aromatic compound composition is missing. Text: … for the purple sweet corn investigated in the present work and those of previously reported corns, …

L388-389: Text: …, thus enhancing properties related to desirable smells. In addition, boiling changes the composition of the volatile compounds through …

L393: In the raw corn juice, 54 compounds were exclusively identified, such as …

L395-396: Did you measure the overall aroma from the juices? If not I would change the text: Thus, a difference of aroma due to the boiling of purple sweet corn juice can be expected.

L399: Text: … (Table 4; Frozen juice)…

L402: Text: In the cited work, severe heat …

L404: Text: … lower temperature (95 °C for 5 min) was applied in the present work. The investigated process conditions could have been insufficient for the release of …

L405-406: Text: The change in anthocyanin content was also in contrast to what is reported in literature, as in general, anthocyanin’s are …

L407-409: Text: However, the present work found that boiling for 5 min increased the total anthocyanin content in purple sweet corn juice (Table 4; Frozen juice).

L415: Table 4: Regarding the anthocyanin content in frozen vs. fresh juice – what are possible explanations for the contradicting results apparently occurring due to the freezing process? How was the samples thawed after freezing (specify in Materials and methods)? The content of the boiled samples seem consistent (thus not being affected by the freezing) while the raw samples seem more ‘sensitive’ to the freezing process. Interesting!

L418-423: Even though enzymatic degradation may occur, thus explain any decrease of anthocyanin’s, the results suggest that freezing (which can be expected to correlate to low enzymatic activities) decrease the anthocyanin content more than that of raw samples. How long, and at which temperatures where the ‘raw’ and ‘frozen’ samples stored for/at? This should be specified in the Materials and Methods.

L426: Text: … contents decreased by …

L428: Text: … fresh juice decreased by …

L430: Text: … phenol content decreased by …

L431: Text: … content increased with …

L431: And consequently antioxidant capacity was decreased by 35.7 and 17.3% after boiling? Is antioxidant activity related the anthocyanin content? It was previously mentioned that the antioxidant content is related to the content of phenols. Please rewrite or provide references or data linking antioxidant activity with anthocyanin content, or compare the chemistry of phenol’s and anthocyanin’s.

L432-433: … after boiling as determined by the DPPH and …

L433: The improvement of antioxidant capacity of fresh raw juice as compared to what? The samples frozen and/or boiled? Which antioxidant capacity? Table 5 states that (P value of 0.012) the freezing not significantly affects (assuming a limit of significance equal to 0.05) the DPPH radical scavenging activity of the samples. I would rewrite L433 to something like: “The high antioxidant capacity observed for fresh raw juice samples (Table 4) seem to correlate with the elevated contents of phenols and anthocyanins”. However, the fact that frozen samples present phenol/anthocyanin content developments where their respective contents either decrease or increase due to the boiling treatment, while for raw samples both contents respond on boiling with a decrease does not justify the degradation statement on L434-436. Please rewrite.

L433: Text: … boiled juices are presented in Fig 2…

L444-445: Text: These results are consistent with previous…

L446: Regarding references: Please be consistent with the format used for the references.

L447: Place figure texts in accordance with author guidelines.

L448: Text: Asterix (*) indicates significant difference between raw and boiled samples…

L451: Please provide reference regarding the statement for gelatinization unless this was confirmed directly by any of the results in the present work.

L456-457: Rewrite the final sentence. Why is it desirable with elevated viscosity and stability? Is it beneficial that the appearance of beverages is homogenous? What about mouthfeel?

L459: Text: This study offer a comprehensive understanding on the…

L461: Was starch gelatinization measured directly? If not rewrite.

L461: Rewrite the statement on fragrance and aromatic compounds. The volatile components identified in the present work is given with a relative content, also, boiling only potentially enhance fragrance related properties as this not has been measured directly. Thus, a concluding sentence on the changes of volatile/aromatic potentially favouring enhanced smell-properties would seem more accurate.

L467-470: Was enzymatic activity in the raw material and the response to heat-blanching tested in the present work? If not remove this from the conclusion (could be moved to discussion).

L471-474: This can be moved to the part covering the fragrance/aromatic section occurring earlier in the conclusion.

Reviewer #2: The study presents the results of original research. The authors investigated the nutritional and physicochemical characteristics of purple sweet corn juice for human consumption, before and after boiling. The results reported do not seem to have been published elsewhere. Experiments, statistics, and other analyses were performed to a high technical standard and the methods were described in sufficient detailed or referenced when already published elsewhere. All data were provided in the manuscript and supported the conclusions. Conclusions were a summary of the main experimental data. One suggestion is that the authors also discuss possible implications for their results in the Conclusions section, especially if boiling of sweet corn juice is beneficial or not to human consumption and nutrition, which seems to be the main goal of this study. The article is presented in an intelligible fashion, is well-structured, and is written in standard English with very few grammatical errors. Please find below some minor corrections and suggestions.

Line, comment/suggestion

36, replace “analyzation” with “analysis”

37, replace “has” with “have”

59, replace “benefit” with “beneficial”

88, insert “of” after “because”

256, insert “, respectively” after Fig. 1

269, replace “B1, B2 and E” with “B1 (VB1), B2 (VB2), and E (VE)” to match the abbreviations in the figure.

271, Table 1, “TAA” line, total% should be 100 not 1.

318, One hypothesis could be that B1 was released from the residue during boiling.

350, delete the parenthesis “)” after “[3,32]”

375, Table 3, units are missing

446, replace “(Mohapatra, Patel, Kar, Deshpande, & Tripathi, 2019)” with correct reference number [16]

130, Are there any references for the following methods or have been developed/modified in house? If reported elsewhere please add the reference and if needed briefly describe the important steps or any modifications. Amino acid profile assay, Fatty acid profile assay, Identification of volatile components, Determination of vitamins B1, B2 and E.

Reviewer #3: General comment: The authors reported the effect of boiling on nutritional and physicochemical properties of purple sweet corn juice. The investigation was appropriate, and the authors did a thorough job in discussing the results (for most part). However, several issues need to be addressed before the manuscript is acceptable for publication.

Specific comments:

Line 28: Change all “antinutrition” to “antinutrient” or “antinutrients”.

Line 30: “stability” is not specific. Stability of what?

Line 30: “Ubbelohde”.

Line 36: Change “analyzation” to “analyses”.

Line 37: “have”.

Line 59: “beneficial”.

Line 69: What does “stability” refer to?

Lines 88-97: Results do not belong here. Write objectives and underlying hypothesis here.

Line 111: “bred”.

Lines 126-129: There should be a separate section for statistical analysis.

Line 232: The Folin-Ciocaulteau method lacks specificity. Why did not the authors analyze anthocyanins and phenolic compounds using an HPLC?

Lines 239-240: Provide detailed experimental procedures.

Line 311: “easily”.

Line 312: “reactive oxygen species”.

Line 333: These fatty acids have much higher boiling point that the tested temperature.

Lines 351-353: Why? Please provide references to support this speculation.

Line 375: What is the unit?

Line 376: “gray background”.

Line 392: “boiled”.

Line 417: Please indicate which comparisons were labeled with lowercase letters for statistical significance and which ones with uppercase letters.

Line 437: This table is unnecessary and should only be included as an appendix.

Lines 440-457: Discussion is very limited in these two sections and should be expanded.

Figure 1: It does not make sense to group sugar and vitamins in one figure.

Figure 2: The antinutrients results can be in one figure, and the physical properties in another.

6. PLOS authors have the option to publish the peer review history of their article (what does this mean?). If published, this will include your full peer review and any attached files.

Reviewer #1: No

Reviewer #2: Yes: Ioannis Dogaris

Reviewer #3: No

---

## [Author Response · Author response to Decision Letter 0]

7 Apr 2020

Reviewer #1

L24-44: The abstract should not include methodological details. Please rephrase/remove. 

Thanks. It has been revised. As the guideline required, we explained how the study was done without methodological detail in the abstract.

L25-28: Text: In order to contribute to the increased knowledge available for consumers on sweet corn juice, the nutritional and physicochemical characteristics of raw and boiled purple sweet juice was herein investigated.

Thanks. It has been revised. Line27-29

L33-34: Refrigerated samples? Is this the freezed or raw samples? Both of those samples reach similar levels on anthocyanin after boiling. Please rephrase. 

Thanks. It has been revised. Line35

L40: Text: The boiling process had no … However, the content of fatty acids were …

Thanks. It has been revised. Line41-43

L42-44: Text: These results provide detailed information relevant for further development of sweet corn juice …

Thanks. It has been revised. Line44-45

L77: Text: Boiling and roasting has been found to…

Thanks. It has been revised. Line78

L92: Is heat-blanching investigated in the present work? If not remove/rephrase. 

Heat-blanching is another expression of boiling processing at 95oC for 5 min. It has been revised. Line92-94

L100-109: CAS numbers for the products and their purity/quality is missing. 

Thanks. There are so many reagents, like amino acids. We don't think it is a good choice to list the CAS number. It will make text verbose. Information of purity has been added in text. Line111-113

L117: What type of juice extractor? Can I expect the same homogenization if I use my extractor at 10 000 rpm for 1 min?

Yes, you can. It is a normal household juicer.

L120: Medical gauze? Details? Pore-size? Material? This will affect the degree of filtration. 

Details have been added. Line124-125

L126-129: Level of significance used (alpha value(s))? As mentioned in the guidelines, this should be specified. 

Details have been added. Line134-136

L132: How is degradation and racemization of amino-acids accounted for in the applied method? Such effects will affect the estimated nutritional values of the juice. 

Maize kernels contain few D-Amino acids. Thus, we do not care about racemization. Hydrolysed amino acids were determined after derivative reaction and quantified by the standard curve. So, the degraded portion could be rectified by the standard curve except for tryptophan which was thoroughly degraded.

L132-134: 1 mL of water was added to the dried/evaporated 1 mL aliquot, then added 1 mL of water before a second evaporation to dryness? This should be written clearer. 

Thanks. It has been revised. Line140

L134: What is the hydrolysate? The original liquid from which 1 mL of supernatant was withdrawn? Or the aliquot eventually evaporated? I guess both are hydrolysates and the difference/similarity should be stated clearer. 

Thanks. It has been revised. Line40

L140: What type of filter paper? Technical details?

Thanks. It has been revised. Line146

L154: Equations should be numbered, referred to in the text, and described in detail. 

Thanks. It has been revised. Line163-164

L164: Ether-petroleum ether mixture? Diethyl ether mixed with petroleum ether? Which ratio?

Details have been added. Line171-172

L171: Added to the extraction and staying at 85 °C? Added to the separated and dried organic phase and kept at 85 °C for 30 min? Please clarify. 

Yes. Details have been added. Line 178

L172: Text: … and kept at 85 …

Thanks. It has been revised. Line 179

L177: Membrane material?

Details have been added. Line184-185

L181: Text: Heating to 200 °C…

Thanks. It has been revised.

L194: Text: Heating to … 

Thanks. It has been revised. Line190

L196: Text: Split ratio = unsplit; inlet temperature = 250 °C; ….

Thanks. It has been revised. 

L219: Text: … to remove the acid.

Thanks. It has been revised. Line233

L224: Text: Detector = DAD-FLD; column temperature = …

Thanks. It has been revised. Line 238-240

L239-240: Please give a short summary of the applied method. 

Summary has been added. Line254-260

L243: Technical details for medical gauze is missing. 

Details have been added. Line263-264

L244: What are the units for the obtained viscosity?

Unit is mm2/s, and it was showed in result.

L247: Why was absorbance used and why 660 nm? Which component in the juice absorbs light at this wavelength and why is this related to the stability of the juice? 

Both absorbance and transmissivity can meet our requirement. For better understanding, absorbance has been replaced by transmissivity in text. Transmissivity at 660 nm was widely used to measure the density of bacterium or small particles in culture medium because the light of this wavelength was blocked mainly by particles in liquid. Before our experiment, long time and high speed centrifuged corn juice was tested under 660 nm, and there was no absorption peak. However, non-centrifuged juice can block the vast majority of light. The more unstable the liquid system, the more rapid the particles deposited during centrifugation. Thus, a moderate speed and time for centrifugation was chose and stability was expressed as a ratio of transmissivity between post- and pre- centrifugation. Line267-269

L271: Table 1: Please clarify (with details, or correct) how there is a statistical difference (significance level of 0.05) between for example Raw vs. Boiled samples and their Methionine content considering the concentration in the juice is given as 297.62 ± 25.64 g/L vs. 298.96 ± 15.80 g/L, respectively. The same holds for Thr, Lys, Phe, Leu, …

I am sorry. I think that you may confuse the information of Fig1 and Table1. There is no difference between raw and boiled sample in Table1.

L280: Text: Casein, a dairy protein…

Thanks. It has been revised. Line302

L596: Figure 1 contain abbreviations not described in the text (VB1, VB2 and VE) and Figure text is missing for both Figure 1 and 2. Caption for Figure 2 is on L447, while caption for Figure 1 is missing. Please follow instructions given in the guidelines. 

I am sorry. I think that you may miss it. It was there before Table1. 

L301-303: I guess the values obtained for sweet and normal field corns are obtained from reference [5]? Anyways, please include the reference subsequent the states values, or at the end of L303. 

Thanks. It has been revised. Line326

L308: Text: … loss of vitamin B1 upon boiling and steaming…

Thanks. It has been revised. Line330

L310: Text: … and only slightly affected by thermal processing.

Thanks. It has been revised. Line332-333

L311: Text: … is easily degraded by light…

Thanks. It has been revised. Line333

L312: Text: … is susceptible for chemical modifications in …

Thanks. It has been revised. Line334

L321: Text: … in the germ, with a fraction of... being as high as…

Thanks. It has been revised. Line343

L325: Text: … 35 different fatty acids were identified (fatty acids with a content less than 0.5 mg/L are not included)…

Thanks. It has been revised. Line347-348

L330-333: Text: Considering the presence of aromatic compounds (Table 3), it was initially speculated that boiling enhanced fatty acid volatility, promoting their evaporation. This because …

Thanks. It has been revised. Line352-354

L335-337: Please avoid personal pronouns. Text: However, it is believed that… What is the volatility of long chained fatty acids? Boiling points or vapour pressure for pure components? Text: Thus, the residues from … were also analysed.

Thanks. It has been revised. Line358-358

L341: Avoid ‘we’. 

Thanks. It has been revised. Line362

L341-344: Very interesting!

Thanks.

L345: Text: On a dry weight basis, …

Thanks. It has been revised. Line366

L358: Three biological replicates of what? Of raw and boiled purple sweet corn juice? Clarify. 

Thanks. It has been revised. Line380

L364: Text: The presence of most compounds varied…

Thanks. It has been revised. Line385-386

L366-367: Text: …, suggesting they occurred in trace amounts. 

Thanks. It has been revised. Line388

L369: Text: … in the purple sweet …

Thanks. It has been revised. Line390

L375: Table 3: Units are missing. Which internal standard was used during GC-MS for fatty acids and volatile components? Please include this information in the appropriate sections within the Materials and methods. 

I am sorry. It is not quantitative data here. It is relative content of each component analyzed by area normalization method. Details have been added to the Materials and methods. Line208-209

L382: Text: … which has been reported to occur persistently [References]. 

Thanks. It has been revised. Line401

L384: Text: … whereas in the present work it was found to be a major component (~10%) within the aromatic compounds.

Thanks. It has been revised. Line403-404

L386: References relating to the statement regarding aromatic compound composition is missing. Text: … for the purple sweet corn investigated in the present work and those of previously reported corns, …

Thanks. It has been revised. Line405-406

L388-389: Text: …, thus enhancing properties related to desirable smells. In addition, boiling changes the composition of the volatile compounds through … 

Thanks. It has been revised. Line409-410

L393: In the raw corn juice, 54 compounds were exclusively identified, such as …

Thanks. It has been revised. Line413-414

L395-396: Did you measure the overall aroma from the juices? If not I would change the text: Thus, a difference of aroma due to the boiling of purple sweet corn juice can be expected. 

Thank you. This change is better. Line415-416

L399: Text: … (Table 4; Frozen juice)…

Thanks. It has been revised. Line421

L402: Text: In the cited work, severe heat …

Thanks. It has been revised. Line424

L404: Text: … lower temperature (95 °C for 5 min) was applied in the present work. The investigated process conditions could have been insufficient for the release of … 

Thanks. It has been revised. Line426-427

L405-406: Text: The change in anthocyanin content was also in contrast to what is reported in literature, as in general, anthocyanin’s are …

Thanks. It has been revised. Line428-429

L407-409: Text: However, the present work found that boiling for 5 min increased the total anthocyanin content in purple sweet corn juice (Table 4; Frozen juice). 

Thanks. It has been revised. Line430-432

L415: Table 4: Regarding the anthocyanin content in frozen vs. fresh juice – what are possible explanations for the contradicting results apparently occurring due to the freezing process? How was the samples thawed after freezing (specify in Materials and methods)? The content of the boiled samples seem consistent (thus not being affected by the freezing) while the raw samples seem more ‘sensitive’ to the freezing process. Interesting!

Yes. It can explain the contradicting results. Details about thawing have been added. Line128-129

L418-423: Even though enzymatic degradation may occur, thus explain any decrease of anthocyanin’s, the results suggest that freezing (which can be expected to correlate to low enzymatic activities) decrease the anthocyanin content more than that of raw samples. How long, and at which temperatures where the ‘raw’ and ‘frozen’ samples stored for/at? This should be specified in the Materials and Methods. 

It is not convenient to point out the storing time, because samples for different experiments were stored for different length of time. The samples used in this experiment were specified in result. Line438-439

L426: Text: … contents decreased by …

Thanks. It has been revised. Line441

L428: Text: … fresh juice decreased by …

Thanks. It has been revised. Line443

L430: Text: … phenol content decreased by … 

Thanks. It has been revised. Line445

L431: Text: … content increased with … 

Thanks. It has been revised. Line445

L431: And consequently antioxidant capacity was decreased by 35.7 and 17.3% after boiling? Is antioxidant activity related the anthocyanin content? It was previously mentioned that the antioxidant content is related to the content of phenols. Please rewrite or provide references or data linking antioxidant activity with anthocyanin content, or compare the chemistry of phenol’s and anthocyanin’s. 

Yes. Previous reports cited in introduction were about yellow or white corn, so the content of phenols mainly determined the antioxidant activity. In the present study, purple corn was used, a large amount of anthocyanins will display big effect on the antioxidant activity.

L432-433: … after boiling as determined by the DPPH and …

Thanks. It has been revised. Line446

L433: The improvement of antioxidant capacity of fresh raw juice as compared to what? The samples frozen and/or boiled? Which antioxidant capacity? Table 5 states that (P value of 0.012) the freezing not significantly affects (assuming a limit of significance equal to 0.05) the DPPH radical scavenging activity of the samples. I would rewrite L433 to something like: “The high antioxidant capacity observed for fresh raw juice samples (Table 4) seem to correlate with the elevated contents of phenols and anthocyanins”. However, the fact that frozen samples present phenol/anthocyanin content developments where their respective contents either decrease or increase due to the boiling treatment, while for raw samples both contents respond on boiling with a decrease does not justify the degradation statement on L434-436. Please rewrite. 

Here, P value of 0.012 is small than 0.05, so the difference is significant. This part has been rewritten. Line447-453

L443: Text: … boiled juices are presented in Fig 2…

Thanks. It has been revised. Line463

L444-445: Text: These results are consistent with previous…

Thanks. It has been revised. Line465

L446: Regarding references: Please be consistent with the format used for the references. 

Thanks. It has been revised. Line466

L447: Place figure texts in accordance with author guidelines. 

Thanks. It has been revised.

L448: Text: Asterix (*) indicates significant difference between raw and boiled samples…

Thanks. It has been revised. Line 469-470

L451: Please provide reference regarding the statement for gelatinization unless this was confirmed directly by any of the results in the present work. 

Reference has been added. Line474

L456-457: Rewrite the final sentence. Why is it desirable with elevated viscosity and stability? Is it beneficial that the appearance of beverages is homogenous? What about mouthfeel? 

Yes. It has been revised. Line477-479

L459: Text: This study offer a comprehensive understanding on the… 

Yes. It has been revised. Line481

L461: Was starch gelatinization measured directly? If not rewrite. 

It has been rewritten. Line483

L461: Rewrite the statement on fragrance and aromatic compounds. The volatile components identified in the present work is given with a relative content, also, boiling only potentially enhance fragrance related properties as this not has been measured directly. Thus, a concluding sentence on the changes of volatile/aromatic potentially favouring enhanced smell-properties would seem more accurate. 

It has been rewritten. Line483

L467-470: Was enzymatic activity in the raw material and the response to heat-blanching tested in the present work? If not remove this from the conclusion (could be moved to discussion). 

Thanks. It has been revised.

L471-474: This can be moved to the part covering the fragrance/aromatic section occurring earlier in the conclusion.

Thanks. It has been revised. Line484-487

Reviewer #2

One suggestion is that the authors also discuss possible implications for their results in the Conclusions section, especially if boiling of sweet corn juice is beneficial or not to human consumption and nutrition, which seems to be the main goal of this study. 

Boiling processing has advantageous and disadvantageous effect on purple sweet corn juice. Here we just contribute to the increased knowledge available for consumers on sweet corn juice. It is not easy to make a decision about which kind of juice is better based on these results.

36, replace “analyzation” with “analysis”

Thanks. It has been revised. Line37

37, replace “has” with “have”

Thanks. It has been revised. Line38

59, replace “benefit” with “beneficial”

Thanks. It has been revised. Line60

88, insert “of” after “because”

Thanks. It has been revised. Line89

256, insert “, respectively” after Fig. 1

Thanks. It has been revised. Line276

269, replace “B1, B2 and E” with “B1 (VB1), B2 (VB2), and E (VE)” to match the abbreviations in the figure.

Thanks. It has been revised. Line295

271, Table 1, “TAA” line, total% should be 100 not 1.

Thanks. It has been revised.

318, One hypothesis could be that B1 was released from the residue during boiling.

However, vitamin B2 decreased from 0.62 to 0.34 mg/L. Thus, I think that VB1 release from the residue during boiling is not a good explanation.

350, delete the parenthesis “)” after “[3,32]”

Thanks. It has been revised.

375, Table 3, units are missing

I am sorry. It is not quantitative data here. It is relative content of each component analyzed by area normalization method.

446, replace “(Mohapatra, Patel, Kar, Deshpande, & Tripathi, 2019)” with correct reference number [16]

Thanks. It has been revised. Line465

130, Are there any references for the following methods or have been developed/modified in house? If reported elsewhere please add the reference and if needed briefly describe the important steps or any modifications. Amino acid profile assay, Fatty acid profile assay, Identification of volatile components, Determination of vitamins B1, B2 and E.

References have been added. Line149, line188, line202, line211, line228

Reviewer #3

Line 28: Change all “antinutrition” to “antinutrient” or “antinutrients”.

Thanks. It has been revised. Line29

Line 30: “stability” is not specific. Stability of what?

Thanks. It has been revised. Line31

Line 30: “Ubbelohde”.

Thanks. It has been revised. Line31

Line 36: Change “analyzation” to “analyses”.

Thanks. It has been revised. Line37

Line 37: “have”.

Thanks. It has been revised. Line38

Line 59: “beneficial”.

Thanks. It has been revised. Line60

Line 69: What does “stability” refer to?

It means that the appearance of beverage is homogenous and sedimentation is slow.

Lines 88-97: Results do not belong here. Write objectives and underlying hypothesis here.

Thanks. We write it based on the guidelines of PlosOne. “The introduction should: Provide background that puts the manuscript into context and allows readers outside the field to understand the purpose and significance of the study; Define the problem addressed and why it is important; Include a brief review of the key literature; Note any relevant controversies or disagreements in the field; Conclude with a brief statement of the overall aim of the work and a comment about whether that aim was achieved.”

Line 111: “bred”.

Thanks. It has been revised. Line115

Lines 126-129: There should be a separate section for statistical analysis.

Thanks. It has been revised. Line131

Line 232: The Folin-Ciocaulteau method lacks specificity. Why did not the authors analyze anthocyanins and phenolic compounds using an HPLC?

In this study, we just want to determinate the total phenols and total anthocyanins. The method used here is simple, economical, and fit for the experimental needs.

Lines 239-240: Provide detailed experimental procedures.

Details have been added. Line254-260

Line 311: “easily”.

Thanks. It has been revised. Line333

Line 312: “reactive oxygen species”.

Thanks. It has been revised. Line334

Line 333: These fatty acids have much higher boiling point that the tested temperature.

It has been rewritten. We think these fatty acids were carried into air by vapour. Line352-354

Lines 351-353: Why? Please provide references to support this speculation.

It has been rewritten. Line372-374

Line 375: What is the unit?

I am sorry. It is not quantitative data here. It is relative content of each component analyzed by area normalization method.

Line 376: “gray background”.

Thanks. It has been revised. Line418

Line 392: “boiled”.

Based on the method, aromatic compounds were collected during boiling, so “boiling” was used here.

Line 417: Please indicate which comparisons were labeled with lowercase letters for statistical significance and which ones with uppercase letters.

Details have been added. Line457

Line 437: This table is unnecessary and should only be included as an appendix.

Thanks. It has been revised.

Lines 440-457: Discussion is very limited in these two sections and should be expanded.

It has been rewritten. Line459-479

Figure 1: It does not make sense to group sugar and vitamins in one figure.

Thanks. We want the picture to be more condensed. Here the vertical line between sugar and vitamin has been modified and is more visible now.

Figure 2: The antinutrients results can be in one figure, and the physical properties in another.

Thanks. We want the picture to be more condensed. We believe it does not interfere with the information transfer.

---

## [Decision Letter · Decision Letter 1]

20 Apr 2020

PONE-D-20-06814R1

Nutritional and physicochemical characteristics of purple sweet corn juice before and after boiling

PLOS ONE

Dear Miss Lu,

Thank you for submitting your manuscript to PLOS ONE. After careful consideration, we feel that it has merit but does not fully meet PLOS ONE’s publication criteria as it currently stands. Therefore, we invite you to submit a revised version of the manuscript that addresses the points raised during the review process.

The quality of your manuscript has been singificantly improved. The reviewers have pointed-our some minor corrections in the text that should be done prion to its acceptance. Please ammend your manuscript accordingly.

We would appreciate receiving your revised manuscript by Jun 04 2020 11:59PM. To enhance the reproducibility of your results, we recommend that if applicable you deposit your laboratory protocols in protocols.io, where a protocol can be assigned its own identifier (DOI) such that it can be cited independently in the future. For instructions see: http://journals.plos.org/plosone/s/submission-guidelines#loc-laboratory-protocols

We look forward to receiving your revised manuscript.

Kind regards,

Leonidas Matsakas

Academic Editor

PLOS ONE

Reviewers' comments:

Reviewer's Responses to Questions

**Comments to the Author**

1. If the authors have adequately addressed your comments raised in a previous round of review and you feel that this manuscript is now acceptable for publication, you may indicate that here to bypass the “Comments to the Author” section, enter your conflict of interest statement in the “Confidential to Editor” section, and submit your "Accept" recommendation.

Reviewer #1: All comments have been addressed

Reviewer #2: All comments have been addressed

Reviewer #3: All comments have been addressed

2. Is the manuscript technically sound, and do the data support the conclusions?

Reviewer #1: Yes

Reviewer #2: (No Response)

Reviewer #3: Yes

3. Has the statistical analysis been performed appropriately and rigorously? 

Reviewer #1: Yes

Reviewer #2: (No Response)

Reviewer #3: Yes

4. Have the authors made all data underlying the findings in their manuscript fully available?

Reviewer #1: Yes

Reviewer #2: (No Response)

Reviewer #3: Yes

5. Is the manuscript presented in an intelligible fashion and written in standard English?

Reviewer #1: Yes

Reviewer #2: (No Response)

Reviewer #3: Yes

6. Review Comments to the Author

Reviewer #1: L38: Text: “Several aromatic compounds not previously reported were identified in sweet purple corn …”.

L113: Text: “Purity level of chemicals used for High-Performance Liquid Chromatography (HPLC) and Gas Chromatography Mass Spectroscopy (GCMS) complied with the requirement of the relevant procedures. All other chemicals were of analytical grade”.

L149-150 and L188-189 and L202-203 and L211-212 and L228-229: Text: “The applied process was modified from the national …”.

Comment to earlier remark on statistics in Table 1: Yes, I misunderstood the presentation. You apply both an asterisk the represent statistical significance and essential/semi-essential amino acids. This could be misleading or confusing (evidently). Different symbols could avoid any chance of misinterpretation.

Reviewer #2: (No Response)

Reviewer #3: General comment: The authors did a thorough job revising the manuscript and addressed all my questions. I have only a few minor comments and suggestions.

Minor comments:

Line 27: This sentence is awkward. Why would consumer need to know about the physicochemical characteristics of sweet corn juice? Promoting consumer awareness of the nutritional value or improving quality of the product would be more reasonable justifications for the study.

Lines 28-29: “sweet corn juices”.

Line 78: “have been”.

Line 113: “were of analytical grade”.

Line 135: “ANOVA”.

Line 256: “by measuring absorbance at 410 nm using a UV-Vis spectrophotometer”.

Line 111: change “transmissivity” to “transmittance”.

7. PLOS authors have the option to publish the peer review history of their article (what does this mean?). If published, this will include your full peer review and any attached files.

Reviewer #1: No

Reviewer #2: Yes: Ioannis Dogaris

Reviewer #3: No

---

## [Author Response · Author response to Decision Letter 1]

27 Apr 2020

Reviewer #1: L38: Text: “Several aromatic compounds not previously reported were identified in sweet purple corn …”.

Thanks. It has been revised.

L113: Text: “Purity level of chemicals used for High-Performance Liquid Chromatography (HPLC) and Gas Chromatography Mass Spectroscopy (GCMS) complied with the requirement of the relevant procedures. All other chemicals were of analytical grade”.

Thanks. It has been revised.

L149-150 and L188-189 and L202-203 and L211-212 and L228-229: Text: “The applied process was modified from the national …”.

Thanks. It has been revised.

Comment to earlier remark on statistics in Table 1: Yes, I misunderstood the presentation. You apply both an asterisk the represent statistical significance and essential/semi-essential amino acids. This could be misleading or confusing (evidently). Different symbols could avoid any chance of misinterpretation.

Thanks. A different symbol has been used in Table 1.

Reviewer #2: (No Response)

Reviewer #3: General comment: The authors did a thorough job revising the manuscript and addressed all my questions. I have only a few minor comments and suggestions.

Minor comments:

Line 27: This sentence is awkward. Why would consumer need to know about the physicochemical characteristics of sweet corn juice? Promoting consumer awareness of the nutritional value or improving quality of the product would be more reasonable justifications for the study.

I quite agree with your opinion. In this study, our original purpose was to compare many physicochemical characteristics between raw and boiled sweet corn juices, and by which provide consumers and producers more knowledge about sweet corn juice, and help consumers to make a decision during sweet corn juice consumption. This sentence has been rewritten.

Lines 28-29: “sweet corn juices”.

Thanks. It has been revised.

Line 78: “have been”.

Thanks. It has been revised.

Line 113: “were of analytical grade”.

Thanks. It has been revised.

Line 135: “ANOVA”.

Thanks. It has been revised.

Line 256: “by measuring absorbance at 410 nm using a UV-Vis spectrophotometer”.

Thanks. It has been revised.

Line 267: change “transmissivity” to “transmittance”.

Thanks. It has been revised.

---

## [Editor Report · Decision Letter 2]

29 Apr 2020

Nutritional and physicochemical characteristics of purple sweet corn juice before and after boiling

PONE-D-20-06814R2

Dear Dr. Lu,

We are pleased to inform you that your manuscript has been judged scientifically suitable for publication and will be formally accepted for publication once it complies with all outstanding technical requirements.

With kind regards,

Leonidas Matsakas

Academic Editor

PLOS ONE
---

## [Editor Report · Acceptance letter]

1 May 2020

PONE-D-20-06814R2 

Nutritional and physicochemical characteristics of purple sweet corn juice before and after boiling 

Dear Dr. Lu:

I am pleased to inform you that your manuscript has been deemed suitable for publication in PLOS ONE. Congratulations! Your manuscript is now with our production department. 

With kind regards,

on behalf of

Dr. Leonidas Matsakas 

Academic Editor

PLOS ONE